# CWATR: Generating Richer Captions with Object Attributes

## Abstract

Image captioning is a popular yet challenging task which is at the intersection of Computer Vision and Natural Language Processing. Recently, transformer-based unified Vision and Language models advanced the state-of-the-art further on image captioning. However, there are still fundamental problems in these models. Even though the generated captions by these models are grammatically correct and describe the input image fairly good, they might overlook important details in the image. In this paper, we demonstrate these problems in a state-of-the-art baseline image captioning method and analyze the reasoning behind these problems. We propose a novel approach, named *CWATR (Captioning With ATtRibutes)*, to integrate object attributes to the generated captions in order to obtain richer and more detailed captions. Our analyses demonstrate that the proposed approach generates richer and more visually grounded captions by integrating attributes of the objects in the scene to the generated captions successfully.

## 1 Introduction

With the recent advancements in Computer Vision (CV) and Natural Language Processing (NLP), machines can understand and respond to visual or textual data, and establish relationships between these two modalities. Among several studies working on these two modalities, image captioning aims to generate grammatically and semantically meaningful sentences describing the given input image like the humans do. A good caption should be grammatically correct, natural sounding, rich, and grounded on the image (Stefanini et al., 2022), (Rohrbach et al., 2018), (Zhou et al., 2020b).

Design of large transformer-based models (Vaswani et al., 2017) and utilization of large datasets (Chen et al., 2015; Sharma et al., 2018; Ordonez et al., 2011), (Young et al., 2014), have led to a significant improvement in state-of-the-art in image captioning. OSCAR (Li et al., 2020) and VIVO (Hu et al., 2021) achieved very good results in general image captioning (Chen et al., 2015) and novel object captioning (Agrawal et al., 2019). VinVL (Zhang et al., 2021) further improved those two and achieved state-of-the-art by utilizing richer regional features.

Even though theoretical evaluation of state-of-the-art methods results in high scores, there are overlooked problems in these models. These problems arise when actual captioning outputs are examined in detail. There are studies (Yang et al., 2019; Ma et al., 2020) demonstrating that image captioning models are inclined towards copying phrases from training dataset without paying attention to the input image. Furthermore, these models might hallucinate non-existing objects in the image or overlook important details (Yang et al., 2019; Rohrbach et al., 2018).

Our observations in this study are also in parallel with those findings. The results show that the recent captioning models overlook some aspects of the scene. Most of the time, the generated captions lack details of objects in the scene. An example of such a case is demonstrated in Figure 1. In this example, the caption generated by VIVO (Visual Vocabulary Pretraining) with VinVL features (Hu et al., 2021; Zhang et al., 2021) hallucinates a *chair*. It also overlooks important details in the image such as the *fence* of the garden and the *car* in the background. Moreover it does not mention about properties of the objects in the scene, such as "*small* garden, *red* car".

In this paper, we attack this problem and propose a novel approach in order to generate richer captions with additional object attribute information. More precisely, contributions of this paper are as follows:

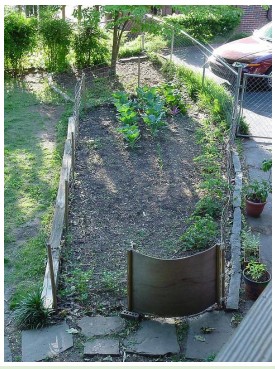

A garden with a chair and a plant on the ground.

Figure 1: An example image and a poor caption generated by VIVO (Hu et al., 2021; Zhang et al., 2021).

- We analyze shortcomings and problems with state-of-the-art image captioning models and demonstrate how they overlook important details in the image.
- We propose a novel approach in order to improve generated captions with object attribute information. The proposed approach, CWATR, is directly applicable to Vision and Language Pretraining (VLP) based approaches.
- We compare the CWATR with the baseline method, VIVO (with VinVL features) (Hu et al., 2021; Zhang et al., 2021), and provide theoretical and visual analyses.

## 2 RELATED WORK

Initial image captioning approaches were based on template filling (Kulkarni et al., 2011; Yao et al., 2010). Predefined sentence templates were filled with predicted object names, attributes, and prepositions. Rapid progress in deep learning field also influenced the image captioning research and deep learning based approaches were proposed (Vinyals et al., 2015; Karpathy & Fei-Fei, 2015). These approaches utilized a Convolutional Neural Network (CNN) as the image encoder and a Recurrent Neural Network (RNN) as the language model.

Later, Xu et al. (2015) proposed *Show, Attend, and Tell* by integrating an attention mechanism between CNN encoder and RNN decoder in order to generate enhanced and more visually grounded captions. After *Show, Attend, and Tell*, attention mechanism became a standard and many other works employed and/or improved attention in image captioning methods (Lu et al., 2017; Chen et al., 2018; 2017).

Anderson et al. (2018) introduced another attention mechanism named Bottom-Up attention in addition to the Top-Down attention in *Show, Attend, and Tell*. In Bottom-Up attention, an object detection algorithm detects objects in the scene. Later, RNN attends to regional features for detected object instead of whole feature map. The idea of exploiting regional features for objects in the scene is employed in many image captioning methods (Qin et al., 2019; Ke et al., 2019; Wang et al., 2020).

The invention of transformers (Vaswani et al., 2017) caused a paradigm shift in NLP and transformers have become the go-to method in recent approaches (Devlin et al., 2019; Radford et al., 2019; Brown et al., 2020). Recently, they become quite popular in CV as well. Transformers are used as a feature extractor similar to CNNs (Dosovitskiy et al., 2021; Touvron et al., 2021). These approaches, dubbed Vision Transformers, divide the image into patches and perform self-attention on these patches. Inspired from these approaches, some image captioning methods employed Vision Transformers in their Visual Encoder blocks (Liu et al., 2021; Wang et al., 2022).

Another branch of image captioning approaches focus on unified architecture for Visual Encoder and Language Model blocks by utilizing transformers. This kind of approach was first introduced in Zhou et al. (2020a), called Unified-VLP (Unified Vision and Language Pretraining). In Unified-VLP, a single transformer network is used for both encoding and decoding steps. It is also unified

in the sense that, a single architecture is pretrained on large image-text paired datasets and then finetuned on task-specific datasets for downstream tasks such as image captioning or visual question answering. OSCAR (Li et al., 2020) further improved Unified-VLP with the integration of object tags which helps alignment between image features and language features. VinVL (Zhang et al., 2021) has been proposed as an improvement over OSCAR. In VinVL, the captioning architecture is the same as OSCAR but the object detection model is improved by training a larger Faster R-CNN (Ren et al., 2015) model on a collection of datasets (Chen et al., 2015; Kuznetsova et al., 2020; Shao et al., 2019; Krishna et al., 2017) instead of just one (Krishna et al., 2017). This helps extracting richer regional features and results in better captioning performance (Stefanini et al., 2022).

VIVO (Visual Vocabulary Pretraining) (Hu et al., 2021) is another Unified-VLP based method which was proposed for Novel Object Captioning where aim is generating captions for images which contain unseen objects during training. In VIVO, pretraining is performed on an object detection dataset (Kuznetsova et al., 2020) and finetuning is performed on an image captioning dataset (Chen et al., 2015) following the restrictions of *nocaps* challenge (Agrawal et al., 2019). VinVL also improves VIVO by just utilizing richer regional features extracted by a larger object detection model while keeping captioning model and strategy the same.

Even though these models generate pretty impressive captions, there are still fundamental problems with them. In Yang et al. (2019), they have shown that image captioning models are prone to biases in the training dataset and language model. They copy phrases from the training set in the generated caption, overlook details in the image and disregard attributes of objects in the image. Various approaches are proposed in order to generate more visually grounded captions (Yang et al., 2019), (Ma et al., 2020). In Wu et al. (2016); Yao et al. (2016), they proposed predicting attributes, embedding them, and feeding them to RNN-decoder as additional input. In Yang et al. (2019), they utilized attributes implicitly by training an attribute prediction module to generate richer features. However, effect of object attributes are unexamined in state-of-the-art unified image captioning models.

In this work, we propose a novel approach for integrating object attributes to the state-of-the-art transformer-based unified image captioning models in order to generate richer captions. We demonstrate the proposed approach on general image captioning and Novel Object Captioning (NOC) tasks. VinVL is the state-of-the-art model on general image captioning and NOC. For NOC, they make use of VIVO pretraining (Hu et al., 2021). Hence, approaches in VIVO and VinVL are taken as baseline in this thesis. Analyses and improvement strategies are employed on this baseline.

## 3 PROPOSED METHOD

We demonstrate our proposed approach, named CWATR (Captioning With ATtRibutes), for generating richer captions with attributes on *nocaps* (Agrawal et al., 2019) challenge. Hence, following VIVO (Hu et al., 2021) and VinVL (Zhang et al., 2021), we pretrain the network on Open Images (Kuznetsova et al., 2020) dataset and finetune on COCO (Chen et al., 2015) dataset.

As the first step in pretraining, finetuning, and testing, we extract regional features, object tags, and object attributes for a given input image, $I$. Regional features and object attributes are extracted using a Faster R-CNN (Ren et al., 2015) model with extra attribute prediction head. This model is the same as in VinVL (Zhang et al., 2021) and it is trained on a collection of datasets (Chen et al., 2015; Kuznetsova et al., 2020; Shao et al., 2019; Krishna et al., 2017). The model is then finetuned on Visual Genome (Krishna et al., 2017) with extra attribute prediction head in order to generate richer regional features. In CWATR, we exploit these attribute predictions.

Regional features are extracted by projecting bounding box predictions of Faster R-CNN to the backbone output feature map and applying RoI pooling.

Faster R-CNN model predicts a set of attributes $M_i = \{m_{i0}, m_{i1}, \ldots, m_{iG_i-1}\}$ for each detected object in the image where $G_i$ is the number of predicted attributes for the $i$-th object. An example attribute prediction output is illustrated in Figure 2. We obtain the set of overall object attributes $A = \{a_0, a_1, \ldots, a_{L-1}\}$ where $L$ is the number of attributes for the input image $I$. We only add maximum of 1 attribute from $M_i$ to the overall set $A$. For an object $o_i$ in $I$, the attribute with the highest confidence in $M_i$ which is also not already existing in the overall set $A$ is added to $A$. Hence, the attributes in $A$ are unique. Such an approach allows extracting a diverse set of most dominant attributes of the objects in the input image.

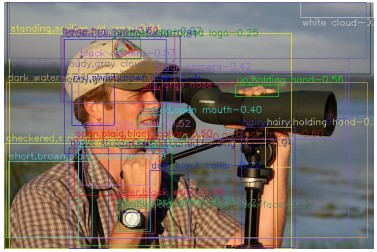

Figure 2: An example object-attribute prediction

Aim in *nocaps* challenge is generating captions for images which contain novel objects that are unseen during captioning training. Models are expected to learn novel objects through Open Images (Kuznetsova et al., 2020) dataset and mention those objects during captioning. Hence, in order to predict novel object tags, we utilize a state-of-the-art multi-label classification network called ML-Decoder (Ridnik et al., 2021) trained on Open Images dataset. This network predicts object tags in the image out of ∼600 novel classes.

The process of extracting regional features, object attributes, and object tags is illustrated is Figure 3. Following this procedure, we obtain 3 sets, $C$, $A$, and $F$, corresponding to object tags, object attributes, and regional features, respectively.

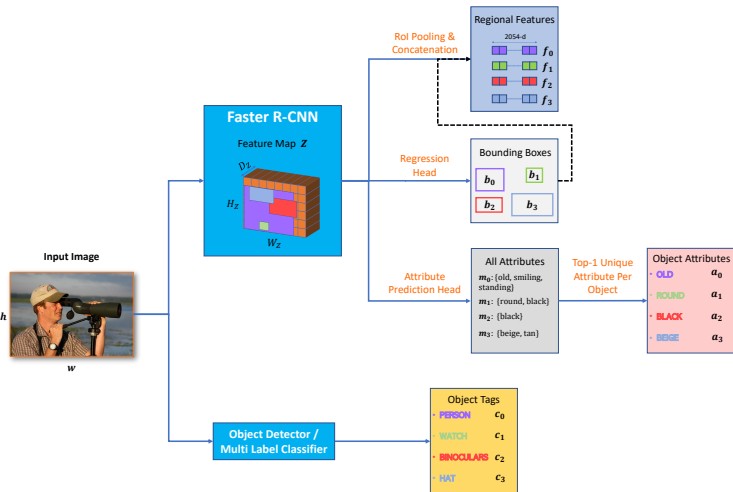

Figure 3: Regional feature, object tag, and object attribute extraction. Regional features and object attributes are extracted using a Faster R-CNN (Ren et al., 2015) model (with extra attribute prediction head) trained on a collection of datasets (Chen et al., 2015; Kuznetsova et al., 2020; Shao et al., 2019; Krishna et al., 2017). Novel object tags are extracted using ML-Decoder (Ridnik et al., 2021) trained on Open Images (Kuznetsova et al., 2020) dataset.

For novel object captioning, we perform pretraining on Open Images (Kuznetsova et al., 2020) dataset, similar to Hu et al. (2021) as the first step. The aim in pretraining is learning a visual vocabulary for novel objects. Different from VIVO, we integrate object attributes to the pretraining process in order to allow model to establish the relationship between regional features, object tags, and object attributes.

During pretraining, previously extracted 3 sets, $C$, $A$, and $F$, are fed to the model. Similar to VIVO, the captioning model is transformer-based $\text{BERT}_{\text{base}}$ (Devlin et al., 2019) network. We randomly replace some of the tokens in object tags (set $C$) and some of the tokens in object attributes (set $A$) with special mask token similar to Devlin et al. (2019). The aim is predicting outputs corresponding

to the masked tokens at the network output by applying self-attention between object tags, object attributes, and regional features.

Formally, we define $\mathcal{D} = \{d_0, d_1, \ldots, d_{M-1}\}$ to be the set of tokens for masked object tags in $C \supseteq \mathcal{D}$ and $\overline{\mathcal{D}} = \{\bar{d}_0, \bar{d}_1, \ldots, \bar{d}_{M-1}\}$ to be the transformer outputs corresponding to elements in $\mathcal{D}$ where M is the number of masked object tags. We define $E = \{e_0, e_1, \ldots, e_{Q-1}\}$ to be the set of tokens for masked object attributes in $A \supseteq E$ and $\overline{E} = \{\bar{e}_0, \bar{e}_1, \ldots, \bar{e}_{Q-1}\}$ to be the transformer outputs corresponding to elements in $E$ where $Q$ is the number of masked attributes.

Input tokens in object tags or object attributes might be fed to the network any order since they are independent. Furthermore, there might be more than one masked token for a set ($M > 1$ or $Q > 1$). In that case, there is an ambiguity at the transformer output since any output for masked token might correspond to any input. As an example, if $round$ and $black$ are two masked input tokens, the network is free to predict these words at any of the outputs since attributes are not ordered by nature unlike words in a sentence. In order to solve this ambiguity, Hu et al. (2021) proposed applying Hungarian Matching between masked words and the network outputs. We employ a similar strategy and perform Hungarian Matching between masked input tokens and corresponding network outputs to obtain an optimal assignment between them.

Since object tags and object attributes belong to different parts of speech, they are treated separately. Two optimal one-to-one assignments are obtained using Hungarian Assignment Algorithm (Kuhn, 1955): $\alpha$ and $\beta$. $\alpha$ assigns masked object tags to transformer outputs for these tags and $\beta$ assigns masked object attributes to transformer outputs for these attributes. To obtain $\alpha$ and $\beta$, Hungarian Assignment Algorithm is solved for two square cost matrices $J^\alpha \in \mathbb{R}^{M \times M}$ and $J^\beta \in \mathbb{R}^{Q \times Q}$, respectively for $\alpha$ and $\beta$. These cost matrices are constructed according to equation 1

$$
\begin{aligned}
J_{ij}^\alpha &= 1 - p(\bar{d}_i^{t_j}) \\
J_{mn}^\beta &= 1 - p(\bar{e}_m^{t_n})
\end{aligned}
\tag{1}
$$

where $p(\bar{d}_i^{t_j})$ and $p(\bar{e}_m^{t_n})$ are the probabilities of transformer outputs $\bar{d}_i$ and $\bar{e}_m$ being the words with vocabulary id $t_j$ and $t_n$, for masked tag and attribute, respectively. $\alpha$ and $\beta$ assigns input indexes to output indexes as in equation 2.

$$
\begin{aligned}
\alpha(i) = j \quad i, j \in [0, M-1] \\
\beta(m) = n \quad m, n \in [0, Q-1]
\end{aligned}
\tag{2}
$$

Hungarian Assignment Algorithm finds $\alpha$ and $\beta$ such that cost functions in equation 3 are minimized.

$$
\begin{aligned}
J_\alpha &= \sum_{i=0}^{M-1} 1 - p(\bar{d}_i^{\alpha(i)}) \\
J_\beta &= \sum_{m=0}^{Q-1} 1 - p(\bar{e}_m^{\alpha(m)})
\end{aligned}
\tag{3}
$$

After the assignments are obtained, the network is trained with Masked Token Loss ($\mathcal{L}_{MTL}$). $\mathcal{L}_{MTL}$ for object tags and object attributes are calculated separately according to equation 4.

$$
\begin{aligned}
\mathcal{L}_{MTL}^{tag} &= -\frac{1}{M} \sum_{d_i \in \mathcal{D}} \sum_{j}^{V} y_i^j \log p(\bar{d}_i^j) \\
\mathcal{L}_{MTL}^{attribute} &= -\frac{1}{Q} \sum_{e_m \in E} \sum_{n}^{V} y_m^n \log p(\bar{e}_m^n)
\end{aligned}
\tag{4}
$$

where $V$ is the size of the vocabulary (number of output classes), $p(\bar{d}_i^j)$ is the predicted output probability for output class $j$ for the masked tag token $d_i$, and $p(\bar{e}_m^n)$ is the predicted output probability for output class $n$ for the masked attribute token $e_m$. $y_i^j$ is the binary ground truth label for class $j$ for the masked word $d_i$ and $y_m^n$ is the binary ground truth label for class $n$ for the masked word $e_m$. $y_i^j$ and $y_m^n$ are calculated using the assignments $\alpha$ and $\beta$ as in equation 5.

$$
y_i^j = \begin{cases} 1, & \text{if } \alpha(i) = j \\ 0, & \text{otherwise} \end{cases}
\qquad
y_m^n = \begin{cases} 1, & \text{if } \beta(m) = n \\ 0, & \text{otherwise} \end{cases}
\tag{5}
$$

Total $\mathcal{L}_{MTL}$ used during pretraining is the sum of individual losses for tags and attributes as in equation 6 where $\lambda$ is used to weight object tag and object attribute losses.

$$\mathcal{L}_{MTL}^{total} = \frac{\mathcal{L}_{MTL}^{tag} + \mathcal{L}_{MTL}^{attribute}}{2} \qquad (6)$$

The process of pretraining is illustrated in Figure 4 for an example input image from Open Images dataset. Full attention mask is used during pretraining. Any input token can attend to any other input token. The pretrained model on Open Images dataset is finetuned on COCO dataset for image

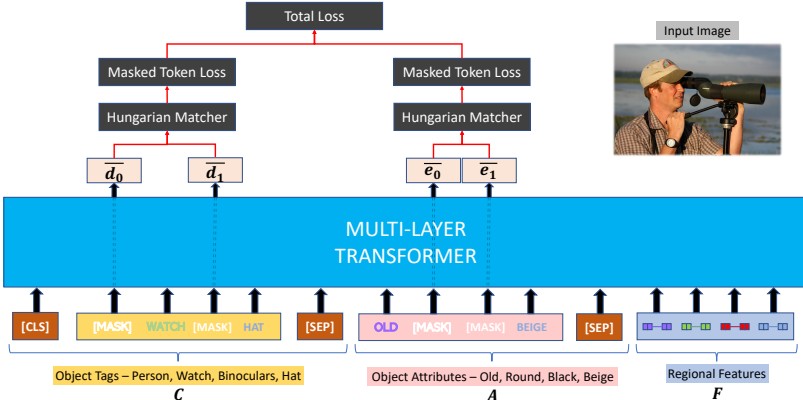

Figure 4: Proposed pretraining approach with object attributes. We feed attributes as an additional set of inputs. Masked object tags and attributes are predicted at the network output by attending to unmasked input vectors from all sets (object tags, object attributes, regional features).

captioning. Firstly, object tags, object attributes, and regional features are extracted in the same way as in pretraining (Figure 3). The set of words in ground truth caption, $S$ is fed to the model along with extracted set of object tags, $C$, object attributes, $A$, and regional features, $F$.

During finetuning, object tags and attributes are not masked. Only tokens in ground truth caption are randomly masked. The model should predict outputs corresponding to the masked words by attending to the other words in the ground truth caption, object tags, object attributes, and regional features. Since the words in ground truth caption are ordered, an assignment is not necessary, unlike pretraining. Formally, let $\mathcal{D} = \{d_0, d_1, \ldots, d_{M-1}\}$ be the set of tokens for masked words in $S \supseteq D$ and $\bar{\mathcal{D}} = \{\bar{d}_0, \bar{d}_1, \ldots, \bar{d}_{M-1}\}$ be the transformer outputs corresponding to elements in $\mathcal{D}$ where $M$ is the number of masked words in $S$. The network is trained with $\mathcal{L}_{MTL}$ objective. Cross-Entropy loss is used in $\mathcal{L}_{MTL}$. $\mathcal{L}_{MTL}$ is calculated according to equation 7.

$$\mathcal{L}_{MTL} = -\frac{1}{M} \sum_{d_i \in \mathcal{D}} \sum_{j}^{V} y_i^j \log p(\bar{d}_i^j) \qquad (7)$$

where $V$ is the size of the vocabulary (number of output classes), $p(\bar{d}_i^j)$ is the predicted output probability for output class $j$ for the masked token $d_i$. $y_i^j$ is the binary ground truth label for class $j$ for the masked word $d_i$.

The process of finetuning is illustrated in Figure 5 for an example input image from COCO dataset. Unidirectional attention mask is used for tokens in the ground truth caption in order to model sequential nature of the language, similar to Hu et al. (2021).

During inference, auto-regressive decoding is applied. At each time instant, input token for that time instant is masked and network is asked to predict the current word at the transformer output by attending to previously predicted words, object tags, object attributes, and regional features.

# 4 EXPERIMENTAL RESULTS

As explained in Section 3, we expose the network to object attributes in both pretraining and finetuning. In this section, we analyze the CWATR model and compare it with the baseline VIVO

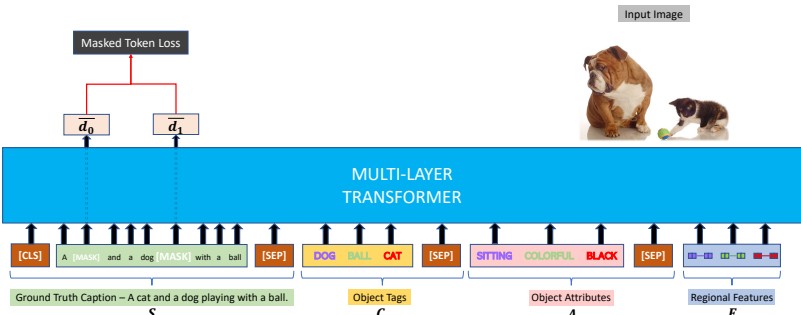

Figure 5: Proposed finetuning approach with object attributes. Attributes are fed as an additional set of inputs. Network is trained with seq2seq objective as in Hu et al. (2021). Masked words in the caption is predicted at the network output by attending to previous words, object tags, object attributes, and regional features.

model which is not exposed to object attributes. Both methods utilize regional features generated by VinVL. We compare two models both visually and theoretically in the following subsections.

## 4.1 VISUAL ANALYSIS

In this section, we provide visual outputs of the baseline VIVO model and the proposed CWATR model. Attribute words in the predicted output caption of the CWATR are shown in bold. Additional visual examples are provided in Appendix A.

In Table 1 and Table 2, we share example images from COCO test set, their ground truth captions, and predicted captions by the VIVO and CWATR.

In Table 1, we see that the CWATR model predicts *small*, *wooden*, and *red* attributes. Furthermore, it mentions about the *car* in the background and does not hallucinate a *chair* object, unlike VIVO. In

Table 1: A visual example, corresponding ground truth sentences and outputs of VIVO and CWATR. CWATR successfully integrates *small, wooden,* and *red* attributes.

| Image | Ground Truth | VIVO | CWATR |
|---|---|---|---|
|  | – A simple, fenced in garden containing several kinds of plants.
– A plot of dirt with a fence around it with flower pots next to it.
– A few plants in a garden near a fence.
– A small garden that has vegetables blooming in it.
– A car some dirt a garden grass bushes and trees. | – A garden with a chair and a plant on the ground. | – A **small** garden with a **wooden** fence and a **red** car behind it. |

Table 2, we observe that the caption generated by the VIVO is very superficial. On the other hand, CWATR produces very detailed description of the scene by mentioning greater number of objects in the scene and integrating an attribute for each of them such as *white, filled, chopped,* and *wooden*. Another visual example, corresponding ground truth sentences and predictions of the models for an

Table 2: A visual example, corresponding ground truth sentences and outputs of VIVO and CWATR. CWATR successfully integrates *white, filled, chopped,* and *wooden* attributes.

| Image | Ground Truth | VIVO | CWATR |
|---|---|---|---|
|  | – A pan filled with veggies and a block of butter.
– Butter or cheese on top of some vegetables in a pan.
– Spoon in a bowl of chopped vegetables with butter.
– A pan with butter and other ingredients for a dish.
– Pot with variety of chopped vegetables, big chunk of butter and spoon. | – A bowl of food with a spoon in it. | – A **white** bowl **filled** with **chopped** veggies and a **wooden** spoon. |

image in *nocaps* validation set is given in Table 3. In this example, VIVO does not give details of

objects in the scene. Furthermore, it hallucinates a *dishwasher* object. On the contrary, CWATR associates *metal* and *wooden* attributes with relevant objects in the scene and does not hallucinate a dishwasher. The success of CWATR in such visual examples urged us to examine visual results

Table 3: A visual example, corresponding ground truth sentences and outputs of VIVO and CWATR. CWATR successfully integrates *metal* and *wooden* attributes.

| Image | Ground Truth | VIVO | CWATR |
|---|---|---|---|
|  | – Wood cabinetry surrounding a sink with a high curved faucet and kitchen knives on the left side of the counter.
– The interior of a kitchen showing the sink and surrounding counter top.
– A kitchen with a cabinet, sink and counter top.
– A clean counter top with a cool cabinet furniture downward.
– A kitchen counter and double sink with a towel laying on the counter.
– Kitchen sink with granite countertop and wooden drawers.
– A sink with knives, plates, and a towel on top of the counter.
– A grey sink is placed in the middle of a countertop.
– A kitchen sink with many cabinets under the sink.
– Knives, dish soap, a plate, an orange container and a towel lie on a kitchen top of the sink. | – A kitchen with a sink, cabinets, and a dishwasher. | – A kitchen with a **metal** sink and **wooden** cabinets. |

further by investigating the ground truth sentences and scores of the models for predicted captions.

In Table 4 and Table 5, we provide examples where CWATR correctly integrates object attributes (*white, sitting, metal, stainless steel*) to the predicted captions but gets lower scores. The reason for these results is that the ground truth captions in these examples describe the scene from different perspectives. For example, in Table 4, some ground truth captions mention about what is written on the poles, some mention about their pointing directions. Some mention about two poles, some mention about only one. Furthermore, they do not mention about some or any of the object attributes. Hence, the proposed CWATR model is penalized for integrating attributes even though the predicted sentences describe the scenes perfectly. Another such example is given in Table 6 where the

Table 4: A visual example, corresponding ground truth sentences and outputs of VIVO and CWATR models. CWATR gets lower score despite producing richer caption.

| Image | Ground Truth | VIVO | CWATR |
|---|---|---|---|
|  | – A street sign that reads " Greta Garbo Strafze ".
– A street sign that reads Greta Garbo Strafe.
– Two intersecting white street signs at an intersection.
– The street sign is reading Greta Garbo on the side of the pole.
– A street sign that is pointing in different ways. | – A close up of a street sign with a sky background.
CIDEr: **61.27**   SPICE: **16.6** | – A couple of **white** street signs **sitting** on top of a **metal** pole.
CIDEr: 38.04   SPICE: 14.29 |

Table 5: A visual example, corresponding ground truth sentences and outputs of VIVO and CWATR models. CWATR gets lower score despite producing richer caption.

| Image | Ground Truth | VIVO | CWATR |
|---|---|---|---|
|  | – We are looking at a small airline toilet.
– A compact sized bathroom with toilet and sink inside.
– A toilet and sink in the bathroom of an airplane.
– A bathroom toilet that has the seat down.
– This is a lavatory of an airplane. | – A bathroom with a toilet, sink, and soap dispenser.
CIDEr: **63.0**   SPICE: **35.71** | – A **white** toilet **sitting** next to a **stainless steel** sink.
CIDEr: 43.64   SPICE: 14.81 |

predictions of the VIVO and CWATR just differs by two attribute words, *white* and *pink*. Despite CWATR identifies and associates these attributes correctly, it gets lower score compared to VIVO. None of the ground truth sentences mention about these attributes. All 5 of them define the scene from different perspectives. Hence, CWATR is punished even though it describes the scene very well and detailed.

## 4.2 THEORETICAL COMPARISON

Comparison of two models on COCO and *nocaps* datasets are given in Table 7 and Table 8, respectively. The results show that the VIVO outperforms the CWATR on both datasets. As discussed in Section 4.1, visual analysis demonstrates that CWATR successfully generates richer and grounded captions with additional attribute information. However it falls behind VIVO because the ground

Table 6: A visual example, corresponding ground truth sentences and outputs of VIVO and CWATR models. CWATR gets lower score despite producing richer caption.

| Image | Ground Truth | VIVO | CWATR |
|---|---|---|---|
| | – Creative centerpiece floral arrangement at an outdoor event.
– A wedding centerpiece made of flowers and various other plants.
– A vase of flowers sitting on an outdoor table.
– A vase filled with flowers on top of a table.
– A floral arrangement inside a clear cylinder shaped vase. | – A vase filled with flowers sitting on top of a table.
CIDEr: **207.57**  SPICE: **33.3** | – A vase filled with **white** and **pink** flowers on top of a table.
CIDEr: 138.99  SPICE: 33.3 |

Table 7: Evaluation results on COCO Karpathy test split (Karpathy & Fei-Fei, 2015) for VIVO baseline model and proposed CWATR model which utilize object attributes.

| Method | COCO | |
|---|---|---|
| | CIDEr | SPICE |
| VIVO | **119.41** | **21.62** |
| CWATR | 110.62 | 20.7 |

Table 8: Evaluation results on *nocaps* validation set for VIVO baseline model and proposed CWATR model which utilize object attributes.

| Method | in-domain | | near-domain | | out-of-domain | | entire | |
|---|---|---|---|---|---|---|---|---|
| | CIDEr | SPICE | CIDEr | SPICE | CIDEr | SPICE | CIDEr | SPICE |
| VIVO | **85.48** | **12.84** | **77.22** | **12.3** | **59.53** | **10.14** | **74.82** | **11.96** |
| CWATR | 79.29 | 12.44 | 72.1 | 11.74 | 55.07 | 9.81 | 69.68 | 11.48 |

truth captions are problematic and do not always contain relevant attributes. Hence, the model which predicts attributes gets penalized even though the generated captions is correct and more detailed compared to the baseline. Additionally, both models' performances are worse on *nocaps* dataset since it is more challenging than COCO dataset due to novel objects.

## 5    CONCLUSION

In this paper, we propose a method, CWATR, to integrate object attributes to state-of-the-art Vision and Language Pretraining based image captioning algorithms in order to generate richer captions. We explain the method in detail. We compare the proposed method with the state-of-the-art baseline method, VIVO (with VinVL features) (Hu et al., 2021; Zhang et al., 2021). Our analysis demonstrates that the proposed method successfully integrates object attributes in the scene to the generated caption. Hence, it generates attribute-wise richer and more detailed captions compared to the baseline method. Further analysis reveals that the proposed method obtains lower scores for some captions compared to the baseline even though the generated captions are better and more detailed than that of the baseline. Our analysis shows that this result is due to improper ground truth captions with high variance. Different people describe the scene at a different level of detail and resulting ground truth captions might lack object attributes. Hence, the proposed method gets penalized in such cases and obtains lower scores. In order to achieve fairer evaluations, we believe that there should be more strict standards while labeling images for captioning (generating ground truth captions). Objects, their attributes, and relationships should be available in the ground truth captions. Such enhanced and consistent captions in the training dataset as well would help training the models better. In addition, the proposed algorithm can be used to guide the process of enhancing ground truth captions. Attribute-wise rich captions predicted by the proposed method can be shown to the annotators as a baseline so that they can originate from these captions and integrate object attributes to the ground truth captions.

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

## A  APPENDIX

Additional visual examples are provided below. Examples in Table 9 and Table 10 are from COCO (Chen et al., 2015) dataset. Examples in Table 11 and Table 12 are from *nocaps* (Agrawal et al., 2019) dataset. Examples in Table 11 and Table 12 reveal the variance in the ground truth captions since they describe the image from very different perspectives.

Table 9: A visual example, corresponding ground truth sentences and outputs of VIVO and CWATR. CWATR successfully integrates *old, rusted, sitting,* and *yellow* attributes.

| Image | Ground Truth | VIVO | CWATR |
|---|---|---|---|
|  | – A old truck with a busted window in the tall bushes.
– A rusty old truck sitting in an overgrown field.
– A rusted out truck parked next to some yellow flowers.
– Old pick-up with flowers growing in front of it.
– A truck is shown decaying among flowers without a window. | – An old truck sitting in the middle of a forest. | – An **old rusted** truck **sitting** in the middle of **yellow** flowers. |

Table 10: A visual example, corresponding ground truth sentences and outputs of VIVO and CWATR. CWATR successfully integrates *double decker* and *parked* attributes.

| Image | Ground Truth | VIVO | CWATR |
|---|---|---|---|
|  | – A tall building with four double decker buses driving along a parking lot.
– A large long bus on a city street.
– Two white double decker buses on a street.
– Double-decker buses sit at the curb in front of an old building.
– Double-Decker buses line up at a bus stop. | – A group of buses that are sitting in the street. | – A group of **double decker** buses **parked** in front of a building. |

Table 11: A visual example, corresponding ground truth sentences and outputs of VIVO and CWATR. CWATR successfully integrates *old* and *white* attributes.

| Image | Ground Truth | VIVO | CWATR |
|---|---|---|---|
|  | – A shabby garage stands next to a brown-red residential building.
– An run down garage, next to a building.
– An old white garage is next to a brown building in front of trees.
– A white garage in disrepair next to a house.
– A garage that is not in a good shape next to a building.
– The large building appears to be a garage.
– A tiny one car garage made of dented sheet metal in the middle of a driveway.
– A garage contains a dented metallic side wall.
– An old white aluminum windowless garage, in disrepair, next to a building.
– A one car garage with galvanized steel outside walls. | – A white door is open outside of a building. | – An **old white** building on the side of a road. |

Table 12: A visual example, corresponding ground truth sentences and outputs of VIVO and CWATR. CWATR successfully integrates *shiny, metal,* and *silver* attributes.

| Image | Ground Truth | VIVO | CWATR |
|---|---|---|---|
|  | – A bronze sculpture of a bust that is shiny.
– With face distorted, the bust in bronze stares ahead.
– A bronze sculpture with a strange expression stands on a brick surface.
– The sculpture of a man made with cooper has the mouth open.
– A bust of a bronzed man is shown with a brick background.
– A statue of a man's face is outside near a brick way.
– A bronze statue of a man who is making a face and has heavy eyebrows.
– A bronze statue of a human face that is not good looking.
– A statue of a man has a bald spot.
– A bronze sculpture of a bearded japanese samurai. | – A statue of a statue of a man's head. | – A close up of a statue of a **shiny metal silver** head. |

