# OpenReview forum: "CWATR: Generating Richer Captions with Object Attributes"
_ICLR.cc/2023/Conference — Submitted to ICLR 2023_

### Official Review · Reviewer_4nF4 · 2022-10-23

**Confidence:** 3
**Correctness:** 2
**Technical Novelty And Significance:** 2
**Empirical Novelty And Significance:** Not applicable
**Recommendation:** 3

**Clarity, Quality, Novelty And Reproducibility:**

The paper is original and well written. The figures are small and not very clear and should be improved.

**Strength And Weaknesses:**

Strengths - The motivation of the paper to generate more descriptive and well grounded captions for an image is well inspired from the limitations of existing methods. They  integrate an additional loss based on object attributes to the pre-training of Vision language transformers.

Weaknesses - #

In the pretraining stage in Fig 4, a very useful ablation / simpler modeling strategy would be to combine the object tags + attributes and then mask that information randomly for token loss prediction.

Due to the absence of masked region modeling, the model might fail to draw associations between attributes / tags and regions and hence fail to generate novel combinations of attributes + tags (black hat - > black dog) during caption generation.

A study of attention maps from the transformers could help to see what muli-modal attention is being learned by the model.

The results are far below the baselines and are not justified well by using human evaluation, I recommend the authors to compare using some human evaluation.

Another metric - SPICE - U [1] could be more useful in measuring such descriptive captions.

The contributions are very limited compared to VIVO (baseline) and the results worse.

[1]Wang, Zeyu, et al. "Towards unique and informative captioning of images." European Conference on Computer Vision. Springer, Cham, 2020.


**Summary Of The Paper:**

The authors propose to address the shortcomings in image captioning models by generating richer captions with additional object attribute information. They pretrain vision language based transformer models using an additional masked attribute loss and match the object information to the corresponding attributes using Hungarian matching algorithm. They further finetune the transformer to generate informative captions conditioned on previous words, object labels , attributes and region based visual features.

**Summary Of The Review:**

The contributions and experimental outcomes of the paper are marginal and not well supported by ablations, human studies or architectural changes. I propose major improvements both in the method and evaluation for a good submission.

---

### Official Review · Reviewer_E9nV · 2022-10-24

**Confidence:** 4
**Correctness:** 3
**Technical Novelty And Significance:** 2
**Empirical Novelty And Significance:** 2
**Recommendation:** 3

**Clarity, Quality, Novelty And Reproducibility:**

The paper is clear enough and we can easily follow it. However, the novelty of the idea and the experimental results are not significant.

**Strength And Weaknesses:**

Strength:

S1. The quality of the generated captions is relatively high.

S2: Using Hungarian Assignment Algorithm in the pre-training stage is interesting to me.

Weaknesses:

W1: The model is not novel at all. In many existing works, people have employed transformers to combine visual features, tags and attributes to enhance vision-language representation, such as Oscar, VIVO, VinVL. And the experimental results are not convincing. The author only compares the proposed model with VIVO. Plus, in the section on visual analysis, the author only describes the differences among the generated captions, but deeper explanations are required. Though the captions generated by the proposed model are more detailed with more adjective words, I think the main reason for this is that the proposed approach employs attributes, while VIVO does not.

W2: It seems that the generated captions are more distinctive, but the author only considers CIDEr and SPICE metrics which are not appropriate for distinctiveness. So I suggest using CIDErBtw [1,2].

[1] J. Wang et al. Compare and reweight: Distinctive image captioning using similar image sets. ECCV, 2020.

[2] J. Wang et al. On Distinctive Image Captioning via Comparing and Reweighting. TPAMI, 2022.

**Summary Of The Paper:**

This paper proposed an image captioning model, termed CWATR, where object labels, attributes, and visual features are combined together using transformers and masked pre-training methods. Basically, the main contribution of this work is that it shows that introducing rich information like visual features, object labels and attributes into image captioning models can improve the quality of the generated captions, such as detailed descriptions.

**Summary Of The Review:**

This paper lacks novelty and convincing experimental results, so I give it a score of 3 in this phase.

---

### Official Review · Reviewer_hWrE · 2022-10-24

**Confidence:** 3
**Correctness:** 2
**Technical Novelty And Significance:** 3
**Empirical Novelty And Significance:** 1
**Recommendation:** 3

**Clarity, Quality, Novelty And Reproducibility:**

Clarity：fairly good with some flaws
Quality：medium
Novelty：fairly good with some flaws
Reproducibility：medium


**Strength And Weaknesses:**

Strength：meaningful attempt in attribute generation.
Weakness: few illustration of the interpretability of experimental methods；the results and conclusion are inadequate and superficial；experiments are insufficient relatively


**Summary Of The Paper:**

The author proposed an approach which concentrate on attributes generation to general image captioning and NOC tasks. It is an interesting attempt, however, there are some serious defects:
1. It is mentioned that existing methods “overlook some aspects of the scene”, it is also the motivation of this work. However, the shortcomings of these methods remain unclear. Methods like [1] also trained on Visual Genome with corresponding attribute. 2. As for the experimental results, there are several examples which CWATR outperforms VIVO. However, it can not prove that it works well on the whole dataset, unless a global metric is designed, like CIDEr. In traditional metrics, VIVO is better than this method.
[1] Xu Yang, Hanwang Zhang, and Jianfei Cai. Learning to Collocate Neural Modules for Image Captioning. 2019 IEEE/CVF International Conference on Computer Vision (ICCV), 2019. doi:10.1109/ICCV.2019.00435


**Summary Of The Review:**

The direction of this paper is innovative, but needs more explanation and design of experimental methods and more filling of experimental results and conclusions. The author could add more experiments to prove the effectiveness of this method.

---

### Decision · Program_Chairs · 2023-01-20

**Decision:**

Reject

**Justification For Why Not Higher Score:**

The reviewers agreed on rejecting the paper and the authors did not provide any feedback.

**Justification For Why Not Lower Score:**

N/A

**Metareview: Summary, Strengths And Weaknesses:**

This paper proposes a new approach for image captioning called CWATR. The method uses object attributes to produce more detailed captions. The reviewers valued the motivation of the paper and the high-level idea of the approach. However, the reviewers have important concerns about the work and they agreed on rejecting the paper. In particular, the reviewers considered the novelty to be very limited, and the experiments to be insufficient and not deep enough. The authors did not provide any feedback.